# In Vitro Efficacy of Flomoxef against Extended-Spectrum Beta-Lactamase-Producing *Escherichia coli* and *Klebsiella pneumoniae* Associated with Urinary Tract Infections in Malaysia

**DOI:** 10.3390/antibiotics10020181

**Published:** 2021-02-11

**Authors:** Soo Tein Ngoi, Cindy Shuan Ju Teh, Chun Wie Chong, Kartini Abdul Jabar, Shiang Chiet Tan, Lean Huat Yu, Kin Chong Leong, Loong Hua Tee, Sazaly AbuBakar

**Affiliations:** 1Department of Medical Microbiology, Faculty of Medicine, University of Malaya, Kuala Lumpur 50603, Malaysia; ngoisootein@um.edu.my (S.T.N.); kartini.abduljabar@ummc.edu.my (K.A.J.); tanshiangchiet@gmail.com (S.C.T.); leoyu_1993@hotmail.com (L.H.Y.); sazaly@um.edu.my (S.A.); 2School of Pharmacy, Monash University Malaysia, Bandar Sunway 47500, Selangor, Malaysia; chong.chunwie@monash.edu; 3Shionogi Singapore Pte Ltd., 10, Anson Road, #34-14 International Plaza, Singapore 079903, Singapore; kinchong.leong@shionogi.com.sg (K.C.L.); loonghua.tee@shionogi.com.sg (L.H.T.); 4Tropical Infectious Diseases Research and Education Centre (TIDREC), University of Malaya, Kuala Lumpur 50603, Malaysia

**Keywords:** antimicrobial resistance, β-lactamase inhibitor, broad-spectrum β-lactamase, resistance gene, *Enterobacteriaceae*

## Abstract

The increasing prevalence of extended-spectrum β-lactamase (ESBL)-producing *Enterobacteriaceae* has greatly affected the clinical efficacy of β-lactam antibiotics in the management of urinary tract infections (UTIs). The limited treatment options have resulted in the increased use of carbapenem. However, flomoxef could be a potential carbapenem-sparing strategy for UTIs caused by ESBL-producers. Here, we compared the in vitro susceptibility of UTI-associated ESBL-producers to flomoxef and established β-lactam antibiotics. Fifty *Escherichia coli* and *Klebsiella pneumoniae* strains isolated from urine samples were subjected to broth microdilution assay, and the presence of ESBL genes was detected by polymerase chain reactions. High rates of resistance to amoxicillin-clavulanate (76–80%), ticarcillin-clavulanate (58–76%), and piperacillin-tazobactam (48–50%) were observed, indicated by high minimum inhibitory concentration (MIC) values (32 µg/mL to 128 µg/mL) for both species. The ESBL genes *bla*_CTX-M_ and *bla*_TEM_ were detected in both *E. coli* (58% and 54%, respectively) and *K. pneumoniae* (88% and 74%, respectively), whereas *bla*_SHV_ was found only in *K. pneumoniae* (94%). Carbapenems remained as the most effective antibiotics against ESBL-producing *E. coli* and *K. pneumoniae* associated with UTIs, followed by flomoxef and cephamycins. In conclusion, flomoxef may be a potential alternative to carbapenem for UTIs caused by ESBL-producers in Malaysia.

## 1. Introduction

Urinary tract infections (UTIs) include infections that affect the urethra (urethritis), urinary bladder (cystitis), or kidneys (pyelonephritis) [1]. UTIs are more prevalent in female subjects, even among individuals with a high risk of infections (e.g., catheterized patients) [2]. Based on disease classification by Infectious Diseases Society of America (IDSA) and National Antimicrobial Guidelines (NAG) Malaysia, uncomplicated UTIs include acute, symptomatic bacterial cystitis and acute pyelonephritis in nonpregnant, premenopausal women without urological abnormalities or comorbidities [3,4]. On the other hand, UTI symptoms in men or the presence of a structural or functional abnormality in the urinary tract in women are considered complicated UTIs [4]. Commonly identified etiologic microorganisms in UTIs include *Escherichia coli*, *Staphylococcus* spp., *Klebsiella pneumoniae*, *Enterobacter* spp., *Proteus mirabilis*, *Enterococcus faecalis*, group B *Streptococcus*, *Pseudomonas aeruginosa*, and *Candida* spp. [5,6]. Uncomplicated UTIs in otherwise healthy women in the community are mainly caused by uropathogenic *E. coli* (80%) [2]. The second most common causative agent of UTIs is *K. pneumoniae*, a highly relevant bacteria in complicated UTIs where patients are predisposed to infections due to underlying healthcare-associated risk factors [6,7].

Once diagnosed with UTIs, finite courses of antibiotics are often prescribed to resolve acute symptoms. The recommended first-line antibiotics for the treatment of acute uncomplicated UTIs include nitrofurantoin, trimethoprim-sulfamethoxazole, fosfomycin trometamol, and pivmecillinam (in regions where it is available), in a treatment course ranging from one to seven days [3,8]. In cases of treatment failure, second-line antibiotics that include fluoroquinolones and β-lactams may be prescribed for three to seven days [8]. Besides, fluoroquinolones and β-lactam antibiotics are recommended for the treatment of acute pyelonephritis and complicated UTIs [3,4]. Given the recurrent nature of UTIs, antibiotic treatments may lead to development of resistance in bacterial pathogens, compromising the effectiveness of successive treatments [2]. Moreover, the wide use of antimicrobial prophylaxis in patients predisposed to UTIs has also contributed to the rise in antimicrobial resistance (AMR) [6,7]. 

The increasing prevalence of AMR among the aetiologic agents of UTIs has been observed since the early 2000 s [5]. Worldwide emergence and prevalence of extended-spectrum β-lactamase (ESBL)-producing *Enterobacteriaceae*, including *E. coli* and *K. pneumoniae*, have become a serious threat to public health [9]. The Institute of Medical Research (IMR), a Ministry of Health (MOH) agency in Malaysia, has been actively monitoring AMR rates in clinical isolates since 2003. To date, 41 hospitals and one public health laboratory have contributed to the national AMR database. AMR surveillance for urine isolates has documented relatively high rates of resistance to β-lactam antibiotics such as ampicillin, amoxicillin-clavulanate, piperacillin-tazobactam, cefepime, cefotaxime, ceftazidime, and cefuroxime (10–68% in *E. coli* and 5–40% in *K. pneumoniae*) and low rates for carbapenem resistance (less than 1% in *E. coli* and less than 5% in *K. pneumoniae*) over the past decade [10]. ESBL-producing *E. coli* and *K. pneumoniae* have emerged and spread in Malaysia [11].

Flomoxef is a broad-spectrum oxacephem antibiotic that was introduced to the medical field in the mid-1980s [12]. The in vitro and clinical efficacy of flomoxef have been proven satisfactory in the treatment of infections caused by both Gram-positive and Gram-negative bacteria, with minimal side effects or unexpected laboratory results [13,14,15]. The use of flomoxef in empiric and definitive therapy has been previously evaluated and was proven to be effective for ESBL-producing *E. coli* bacteremia [16]. Preclinical studies in China, Japan, Korea, and Taiwan have reported that flomoxef susceptibility among ESBL-producing *Enterobacteriaceae*, especially in *E. coli*, was comparable to cefmetazole, cefoxitin, imipenem, and meropenem [17,18,19,20,21]. Clinical studies conducted in pediatric patients with UTIs caused by ESBL-producing *E. coli* showed comparable susceptibility between flomoxef, cefmetazole, and imipenem [22,23].

The in vitro activity of flomoxef on ESBL-producing *Enterobacteriaceae* beyond Northeast Asian region has not been investigated. The current literature showed that flomoxef may be a potential alternative to carbapenems for the treatment of UTIs caused by ESBL-producers. Hence, we aimed to investigate the in vitro efficacy of flomoxef in comparison with other established β-lactam antibiotics against ESBL-producing *E. coli* and *K. pneumoniae* associated with UTIs in Malaysia.

## 2. Results

### 2.1. Patient’s Demographics and Bacterial Strain Antimicrobial Susceptibility

Approximately half of the UTI-associated ESBL-producing *E. coli* (*n* = 28; 56%) and *K. pneumoniae* (*n* = 25; 50%) strains occurred in elderly patients (aged 60 and above). Thirty-two percent of *E. coli* and 36% of *K. pneumoniae* strains were isolated from adult patients aged between 18 and 59 years old. The remaining *E. coli* (12%) and *K. pneumoniae* (14%) strains were isolated from patients aged less than 18 years old. Sixty-eight percent of *E. coli* and 52% of *K. pneumoniae* strains were isolated from female patients. AMR data of the bacterial strains for cefuroxime, nitrofurantoin, trimethoprim-sulfamethoxazole, and ciprofloxacin were obtained from hospital diagnostic laboratory and summarized in Table 1.

### 2.2. β-Lactam Resistance in E. coli

Table 2 summarizes both the minimum inhibitory concentration (MIC) value and the corresponding phenotype of the ESBL-producing *E. coli* strains in this study. The MIC range, median (MIC_50_), and 90% efficacy (MIC_90_) values for all tested β-lactam antibiotics are also shown. Briefly, all *E. coli* strains (*n* = 50) were confirmed as ESBL-producers based on the Clinical and Laboratory Standards Institute (CLSI) guidelines. High MIC values (MIC_50_ ≥ 64 μg/mL) were recorded for penicillins, third- and fourth-generation cephalosporins, and two out of five of the β-lactam combination agents when tested against ESBL-producing *E. coli* strains (Table 2). Carbapenems and flomoxef showed the lowest MIC_50_ values (≤ 0.5 μg/mL). In addition, most of the *E. coli* strains (n = 41; 82%) showed simultaneous resistance to at least 10 β-lactam antibiotics (Table 3).

### 2.3. β-Lactam Resistance in K. pneumoniae

All *K. pneumoniae* strains (*n* = 50) were confirmed to be ESBL-producers based on the CLSI guidelines. Similar to the *E. coli* strains examined in this study, high MIC values (MIC_50_ ≥ 64 µg/mL) were recorded for penicillins, β-lactam combination agents (except ceftazidime-clavulanate and cefotaxime-clavulanate), and third- and fourth-generation cephalosporins (Table 4). Carbapenems and flomoxef showed the lowest MIC values (MIC_50_ ≤ 1 µg/mL), although slightly increased compared to that of the *E. coli* strains. In general, *K. pneumoniae* strains showed comparable AMR phenotypes to *E. coli*. Most of the *K. pneumoniae* strains (90%) were simultaneously resistant to at least 10 β-lactam antibiotics (Table 5).

### 2.4. ESBL Genes Detected in E. coli and K. pneumoniae Strains

The ESBL genes *bla*_CTX-M_ and *bla*_TEM_ were commonly detected among the UTI-associated *E. coli* (58% and 54%) and *K. pneumoniae* (88% and 74%) strains examined in this study (Table 6). The *bla*_SHV_ gene was detected only in *K. pneumoniae* strains (94%), while the *bla*_FOX_ gene was absent in both the *E. coli* and *K. pneumoniae* strains. The presence of at least two ESBL genes was more commonly seen in *K. pneumoniae* (86%) than in *E. coli* (34%) strains.

### 2.5. Flomoxef Activity in ESBL-Producing E. coli and K. pneumoniae

Flomoxef effectively inhibits most of the *E. coli* strains (78%) and approximately half of the *K. pneumoniae* strains (56%) at lower concentrations (≤ 1 µg/mL). Ten percent of *E. coli* strains and 30% of *K. pneumoniae* strains were inhibited by a higher flomoxef concentration (2–4 µg/mL). A potentially non-susceptible phenotype (MIC ≥ 8 µg/mL) to flomoxef was observed in both the *E. coli* (12%) and *K. pneumoniae* (14%) strains. The relative efficiency of flomoxef in comparison with imipenem, meropenem, and cefmetazole is shown in Figure 1. Simultaneous susceptibility to the selected antimicrobial agents was common in both *E. coli* and *K. pneumoniae*, indicated by clusters formed at log_10_ MIC ≤ 1 in the scatter plots. Generally speaking, the MICs of carbapenems remained consistently low regardless of the increasing resistance to flomoxef. A positive relationship was inferred for the susceptibility of *E. coli* and *K. pneumoniae* to cefmetazole and flomoxef, as shown by upward trendlines in the scatter plots.

## 3. Discussion

We observed high rates of resistance to penicillin antibiotics, third- and fourth-generation cephalosporins, and a high rate of nonsusceptibility to β-lactam combination agents among the ESBL-producing *E. coli* and *K. pneumoniae* strains isolated from urine samples. Both organisms showed similar AMR trends when tested with β-lactam antibiotics. The multi-resistance patterns detected in the *E. coli* and *K. pneumoniae* strains in this study were accompanied, in most cases, by ESBL-encoding genes. All ESBL-producing strains examined in this study were highly susceptible to carbapenems with low MIC values. Similarly, flomoxef exhibited an inhibitory effect on the ESBL producers at low concentrations, especially in *E. coli*.

The high rate of resistance to penicillins, cephalosporins (except cephamycins), trimethoprim-sulfamethoxazole, and ciprofloxacin observed in this study suggested that these antibiotics might be ineffective in the treatment of UTIs caused by ESBL-producing *E. coli* and *K. pneumoniae* in this region. These antibiotics are commonly prescribed in Malaysian primary care settings, constituting up to 93% of antibiotics prescribed for UTI treatment [24]. β-lactam antibiotics, including penicillins, cephalosporins, and penicillin combinations with an enzyme inhibitor, are the most prescribed antibiotics in both public hospitals and private practice in Malaysia, accounting for approximately 66% of all antibiotics prescribed in primary care settings [25]. The overuse of these antibiotics could have contributed to the development of resistance among *E. coli* and *K. pneumoniae* strains circulating in Malaysia. The NAG recommended an empirical drug for uncomplicated cystitis, nitrofurantoin, which remained highly effective against ESBL-producing *E. coli* and, to a lesser extent, *K. pneumoniae* in our study [4]. Amoxicillin-clavulanate is recommended as an alternative drug for the treatment of uncomplicated cystitis and a drug of choice for pyelonephritis and complicated UTIs in Malaysia [4]. However, high rates of resistance to amoxicillin-clavulanate were observed in ESBL-producing *E. coli* and *K. pneumoniae* in this study. Therefore, obtaining a urine sample for culture and susceptibility testing before starting treatment is essential to avoid treatment failure in infections caused by ESBL-producers.

Combining our findings with that of the national surveillance data, we observed that a greater resistance to penicillins, cephalosporins, and β-lactam combination agents is probably common among ESBL-producing *E. coli* and *K. pneumoniae* associated with UTIs, and carbapenems remained the most effective therapeutic agents in this region [10]. Comparing to in vitro studies in China and Korea, which also examined a wide array of different β-lactam antibiotics, a similar AMR trend was observed among the ESBL producers [19,21]. ESBL-producing *E. coli* and *K. pneumoniae* reported in these studies were generally highly resistant to cephalosporins (MIC_50/90_ ≥ 8/256 µg/mL) but also remained highly susceptible to carbapenems (MIC_50/90_ ≤ 0.125/0.25 µg/mL) and flomoxef (0.06/0.25 µg/mL ≤ MIC_50/90_ ≤ 1/16 µg/mL), as revealed in this study. Similarly, cephamycins remained largely effective against ESBL-producing strains [19,21]. However, ESBL producers in the reported studies showed greater susceptibility to β-lactam combinations, such as amoxicillin-clavulanate (MIC_50/90_ = 8/16 µg/mL) and piperacillin-tazobactam (2/64 µg/mL ≤ MIC_50/90_ ≤ 8/128 µg/mL), compared to the strains examined in this study. Although piperacillin-tazobactam and cefepime have shown comparable efficacy as carbapenems for infections caused by ESBL producers, this use is only recommended when the organisms are tested susceptible to these antibiotics [26]. The highly resistant phenotype of the local ESBL producers seen in this study suggested against the use of piperacillin-tazobactam and cefepime as potential carbapenem-sparing strategies in Malaysia. Cephamycins may be an alternative to carbapenems for the treatment of UTIs caused by ESBL producers [26]. Based on the data obtained in this study, cefmetazole would be a better option compared to cefoxitin, since a greater susceptibility to cefmetazole was seen with the ESBL-producing strains.

A comparison of the MIC_50/90_ values obtained in this study showed that flomoxef exhibited a higher in vitro efficacy against ESBL-producing *E. coli* and *K. pneumoniae* than most of the β-lactam antibiotics tested, except for carbapenems. In the present study, a reduced susceptibility to flomoxef was found associated with an increased resistance to cefmetazole, while carbapenems remained consistently effective against all bacterial strains. Our results showed that carbapenems had the highest activity against the ESBL-producing strains, with the lowest MIC_50/90_ values (≤0.03/0.06–1/1 µg/mL). On the other hand, the MIC_50/90_ of flomoxef were lower than cefmetazole and cefoxitin in both *E. coli* (two- to sixteen-fold reductions) and *K. pneumoniae* (two- to four-fold reductions), indicating a slightly higher in vitro efficacy of flomoxef against the ESBL producers. Consistent observations were made in studies conducted in China, Japan, and Korea, where ESBL producers isolated in these regions were most susceptible to carbapenems, followed by flomoxef and cephamycins (cefmetazole or cefoxitin) [17,19,21]. Clinical studies investigating the use of flomoxef for the treatment of UTIs caused by ESBL-producing *E. coli* showed promising results and comparable efficacy with carbapenems [22,23]. However, a more variable outcome was produced when used in the treatment of bloodstream infections caused by ESBL producers [26]. Taiwanese studies have documented increased mortality with the use of flomoxef in hemodialysis access-related bacteremia and lower efficacy compared to carbapenems for isolates with flomoxef MIC of 2–8 µg/mL [27,28]. Based on the MIC data obtained in this study, flomoxef may be considered as a potential alternative to carbapenems for less severe infections, such as UTIs caused by ESBL-producing *E. coli* and *K. pneumoniae* in this region.

Most of the UTI-associated *E. coli* and *K. pneumoniae* strains examined in this study carried at least one gene encoding for class A β-lactamases, which are known to confer a resistance to penicillins, extended-spectrum cephalosporins, and monobactams [29]. The majority of the strains carried two to three ESBL genes, likely accounting for the high MIC values of β-lactam antibiotics observed in this study. Moreover, similar ESBL gene combinations could have explained the common β-lactam-resistant phenotypes shared between the *E. coli* and *K. pneumoniae* strains. Our findings are in agreement with local studies that identified *bla*_CTX-M_ as the most prevalent gene among the ESBL-producing *E. coli* and *K. pneumoniae* isolated from human and animal sources [30,31,32,33]. In these local reports, ESBL producers often carried multiple ESBL genes conferring a wide range of β-lactam resistance in the organisms, which is consistently observed in this study. The absence of *bla*_FOX_ within our strains pool is probably not unexpected, given that the rates of resistance to cephamycins (cefmetazole and cefoxitin) were relatively low among *E. coli* and *K. pneumoniae.* Consistent with the previous report, the cephamycins remained active against the majority of the investigated strains, as these organisms mostly produced TEM- and/or SHV-type β-lactamases that are ineffective against cephamycins [34]. Approximately one-fifth of the ESBL-producing *E. coli* did not harbor any of the four ESBL genes that were examined in this study. Since porin- and efflux-mediated mechanisms are not common in *E. coli* and often only confer resistance to antibiotics at low concentrations, other ESBL genes (*bla*_OXA_, *bla*_PER_, *bla*_GES_, etc.) could be responsible for the ESBL-producing phenotype of these strains [35,36]. Moreover, the production of AmpC β-lactamases, especially among *E. coli*, is frequently associated with positive ESBL test outcomes [37,38]. 

One of the limitations of this study was the low number of ESBL-producing *E. coli* and *K. pneumoniae* collected within one year from a single study center in Malaysia. Although our strain pools may not represent the entire ESBL-producing *Enterobacteriaceae* population in this region, both temporally and geographically, the data generated in this study provided an insight into the extent of β-lactam resistance among ESBL producers associated with UTIs in Malaysia. Moreover, the susceptibility data obtained in this study support further consideration on the potential use of flomoxef as an alternative treatment for UTIs caused by ESBL producers. Another limitation of the study was that only four β-lactamase genes were targeted; therefore, the data obtained did not provide a comprehensive molecular epidemiology of ESBL production among the UTI-associated *E. coli* and *K. pneumoniae* strains in this region.

## 4. Materials and Methods

### 4.1. Study Site and Patient Data Collection

Bacterial strains collection was conducted at the University Malaya Medical Centre (UMMC). UMMC is a tertiary hospital with a total of 1623 beds, located at the capital city Kuala Lumpur in Malaysia. Basic demographic data of patients were retrieved from the Patient Information Department (previously known as the Medical Record Unit) of UMMC.

### 4.2. Bacterial Strains Collection

A total of 100 ESBL producers, comprising of 50 *E. coli* and 50 *K. pneumoniae* strains, were obtained from the UMMC diagnostic laboratory’s strain collections. All strains were isolated from urine samples of patients admitted to UMMC in 2015 (January–November). The isolation and identification of the bacterial strains, along with the detection of ESBL production, were part of the routine microbiological examination procedures in the hospital’s diagnostic laboratory. Only ESBL-producing *E. coli* and *K. pneumoniae* were selected for this study. Other bacterial species and nonproducers of ESBL isolated from urine samples were excluded from the study. All selected ESBL-producing strains (*n* = 100) were confirmed as the first isolate from urine samples, and only one strain was collected per patient. Bacterial strain collections were terminated upon reaching the targeted number of strains (50 strains for each species). *E. coli* and *K. pneumoniae* strains were further confirmed by Polymerase Chain Reaction (PCR) through the detection of *phoA* and *mdh* genes, respectively [39,40]. AMR data of the bacterial strains pertaining to commonly prescribed antibiotics for UTIs (cefuroxime, ciprofloxacin, nitrofurantoin, and trimethoprim-sulfamethoxazole) were retrieved from the diagnostic laboratory’s database. These four antibiotics were not included in the broth microdilution assays performed in this study.

### 4.3. Broth Microdilution Assays 

The antimicrobial susceptibility of the *E. coli* and *K. pneumoniae* strains to β-lactam antibiotics, including flomoxef, was determined. MIC of penicillins (ampicillin, penicillin-G, piperacillin, and ticarcillin); β-lactam combination agents (amoxicillin-clavulanate, ceftazidime-clavulanate, cefotaxime-clavulanate, ticarcillin-clavulanate, and piperacillin-tazobactam); cephems (cefmetazole, cefoxitin, ceftazidime, cefoperazone, ceftriaxone, cefotaxime, and cefepime); carbapenems (imipenem and meropenem); and flomoxef were determined using the broth microdilution method. The MIC values were interpreted according to the breakpoints recommended by the CLSI guidelines [41]. ESBL-producing phenotype of the bacterial strains was confirmed by at least 8-fold reduction in the MIC of ceftazidime with 4-µg/mL clavulanate (fixed dose) when compared to the MIC of ceftazidime alone or cefotaxime-clavulanate (4-µg/mL fixed dose) when compared to cefotaxime [41]. CLSI-recommended threshold for flomoxef was not available; therefore, the MIC breakpoints for moxalactam (an oxacephem antibiotic) were used to predict the bacterial susceptibility to flomoxef [21]. Scatter plots were constructed to illustrate the relative efficacy of flomoxef in comparison with the β-lactam antibiotics that remained active against ESBL-producing *E. coli* and *K. pneumoniae* strains in this study. The log_10_ MIC values of the comparator antibiotics were plotted against that of flomoxef.

### 4.4. Detection of β-Lactamase-Encoding Genes

PCR was performed to detect the presence of ESBL genes (*bla*_CTX-M_, *bla*_SHV_, and *bla*_TEM_) and the plasmid-borne AmpC-type β-lactamase gene (*bla*_FOX_) in the bacterial strains. Primer sequences, PCR reaction mixture, and thermocycling conditions were adapted from published studies [42,43]. The annealing temperatures of the primers were optimized at 55 °C (*bla*_CTX-M_), 58 °C (*bla*_SHV_ and *bla*_TEM_), and 60 °C (*bla*_FOX_).

## 5. Conclusions

In this study, ESBL-producing *E. coli* and *K. pneumoniae* associated with UTIs in Malaysia were found to be highly resistant to penicillins, third- and fourth-generation cephalosporins, and β-lactam combination agents. Carbapenems showed the highest in vitro efficacy to ESBL-producing strains, followed by flomoxef. Therefore, due to the low prevalence of resistance to flomoxef detected here, this antibiotic can be considered as an alternative to carbapenems in clinical practice in Malaysia for the treatment of UTIs caused by ESBL-producing *E. coli* and *K. pneumoniae*. We recommend the continuous surveillance of ESBL producers at both the institutional and national levels to preserve the clinical efficacy of broad-spectrum cephalosporins and prevent the emergence of carbapenem resistance in local *Enterobacteriaceae* strains.

## Figures and Tables

**Figure 1 antibiotics-10-00181-f001:**
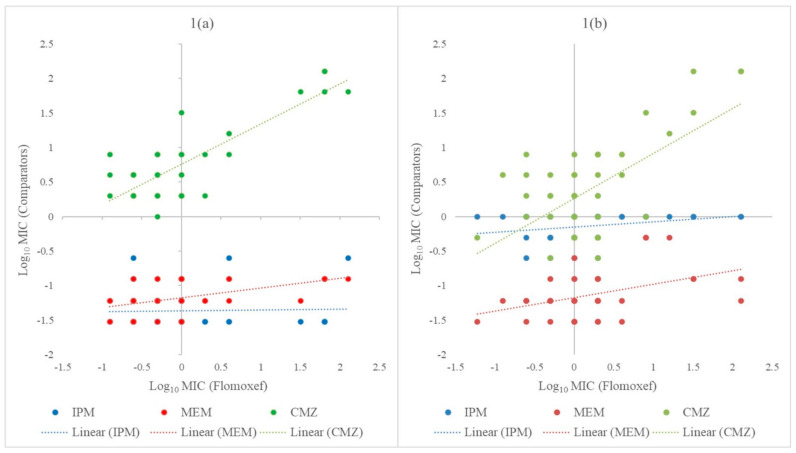
Scatter plots to show the relationship between the minimum inhibitory concentration (MIC) values of imipenem, meropenem, and cefmetazole against flomoxef for *E. coli* (**1a**) and *K. pneumoniae* (**1b**). The dotted lines indicate the linear relationship between the comparators (imipenem, meropenem, or cefmetazole) and flomoxef. CMZ: cefmetazole, IPM: imipenem, MEM: meropenem, and MIC: minimum inhibitory concentration.

**Table 1 antibiotics-10-00181-t001:** Antimicrobial resistance data obtained from the University Malaya Medical Centre (UMMC) diagnostic laboratory database.

Antibiotic	*E. coli* (*n*) (%)	*K. pneumoniae ** (*n*) (%)
Cefuroxime	50 (100)	49 (98)
Ciprofloxacin	35 (70)	20 (40)
Nitrofurantoin	1 (2)	20 (40)
Trimethoprim-sulfamethoxazole	38 (76)	20 (45)

* Nitrofurantoin resistance data for *Klebsiella pneumoniae* was only available for 44 strains. *E. coli*: *Escherichia coli.*

**Table 2 antibiotics-10-00181-t002:** Summary of the minimum inhibitory concentration (MIC) data and susceptibility phenotypes of extended-spectrum β-lactamase (ESBL)-producing *E. coli* strains (*n* = 50).

Antimicrobial Agent	MIC Range (µg/mL)	MIC_50_ (µg/mL)	MIC_90_ (µg/mL)	Susceptibility Phenotype ^a^ (*n*) (%)		
				S	I	R
Ampicillin	128–> 256	>256	>256	0 (0)	0 (0)	50 (100)
Penicillin-G	>64	>64	>64	0 (0)	0 (0)	50 (100)
Piperacillin	≥128	>128	>128	0 (0)	0 (0)	50 (100)
Ticarcillin	>128	>128	>128	0 (0)	0 (0)	50 (100)
Amoxicillin-clavulanate	8–> 64	32	>64	6 (12)	6 (12)	38 (76)
Ceftazidime-clavulanate	1–128	8	32	NA	NA	NA
Cefotaxime-clavulanate	≤0.125–64	8	32	NA	NA	NA
Ticarcillin-clavulanate	8–> 128	128	>128	2 (4)	19 (38)	29 (58)
Piperacillin-tazobactam	4–> 128	64	>128	4 (8)	22 (44)	24 (48)
Cefmetazole	1–> 64	4	64	43 (86)	1 (2)	6 (12)
Cefoxitin	2–> 64	8	>64	29 (58)	10 (20)	11 (12)
Ceftazidime	8–> 256	64	256	0 (0)	1 (2)	49 (98)
Cefoperazone	≥64	>64	>64	0 (0)	0 (0)	50 (100)
Ceftriaxone	32–> 64	>64	>64	0 (0)	0 (0)	50 (100)
Cefotaxime	4–> 256	256	>256	0 (0)	0 (0)	50 (100)
Cefepime	≤0.125–> 256	>256	>256	4 (8)	2 (4)	44 (88)
Imipenem	≤0.03–0.25	≤0.03	0.06	50 (100)	0 (0)	0 (0)
Meropenem	≤0.03–0.125	0.06	0.125	50 (100)	0 (0)	0 (0)
Flomoxef ^b^	≤0.125–> 64	0.5	32	44 (88)	5 (10)	1 (2)

^a^ S: susceptible, I: intermediate, R: resistant, and NA: not available (MIC breakpoints are not provided in the Clinical and Laboratory Standards Institute (CLSI) guidelines). ^b^ Susceptibility phenotypes predicted based on MIC breakpoints for moxalactam (CLSI).

**Table 3 antibiotics-10-00181-t003:** Antimicrobial resistance profiles of *E. coli* (*n* = 50).

AMR Profiles *	No. (%) of Strains
AMP, PCG, PIP, TIC, AMC, TIM, TZP, CMZ, FOX, CAZ, CFP, CRO, CTX, FEP	3 (6)
AMP, PCG, PIP, TIC, AMC, TIM, TZP, FOX, CAZ, CFP, CRO, CTX, FEP	2 (4)
AMP, PCG, PIP, TIC, AMC, TZP, CMZ, FOX, CAZ, CFP, CRO, CTX, FEP	1 (2)
AMP, PCG, PIP, TIC, AMC, TIM, CMZ, FOX, CAZ, CFP, CRO, CTX	1 (2)
AMP, PCG, PIP, TIC, AMC, TIM, FOX, CAZ, CFP, CRO, CTX, FEP	2 (4)
AMP, PCG, PIP, TIC, AMC, TIM, TZP, CAZ, CFP, CRO, CTX, FEP	13 (26)
AMP, PCG, PIP, TIC, AMC, TIM, TZP, FOX, CAZ, CFP, CRO, CTX	1 (2)
AMP, PCG, PIP, TIC, AMC, CMZ, FOX, CAZ, CFP, CRO, CTX	1 (2)
AMP, PCG, PIP, TIC, AMC, TIM, CAZ, CFP, CRO, CTX, FEP	5 (10)
AMP, PCG, PIP, TIC, AMC, TZP, CAZ, CFP, CRO, CTX, FEP	1 (2)
AMP, PCG, PIP, TIC, AMC, CAZ, CFP, CRO, CTX, FEP	6 (12)
AMP, PCG, PIP, TIC, AMC, TIM, CAZ, CFP, CRO, CTX	1 (2)
AMP, PCG, PIP, TIC, TIM, CAZ, CFP, CRO, CTX, FEP	1 (2)
AMP, PCG, PIP, TIC, TZP, CAZ, CFP, CRO, CTX, FEP	3 (6)
AMP, PCG, PIP, TIC, AMC, CAZ, CFP, CRO, CTX	1 (2)
AMP, PCG, PIP, TIC, CAZ, CFP, CRO, CTX, FEP	7 (14)
AMP, PCG, PIP, TIC, CFP, CRO, CTX	1 (2)

* AMP: ampicillin, PCG: penicillin-G, PIP: piperacillin, TIC: ticarcillin, AMC: amoxicillin-clavulanate, TIM: ticarcillin-clavulanate, TZP: piperacillin-tazobactam, CMZ: cefmetazole, FOX: cefoxitin, CAZ: ceftazidime, CFP: cefoperazone, CRO: ceftriaxone, CTX: cefotaxime, and FEP: cefepime.

**Table 4 antibiotics-10-00181-t004:** Summary of the MIC data and susceptibility phenotypes of ESBL-producing *K. pneumoniae* strains (*n* = 50).

Antimicrobial Agent	MIC Range (µg/mL)	MIC_50_ (µg/mL)	MIC_90_ (µg/mL)	Susceptibility Phenotype ^a^ (n) (%)		
				S	I	R
Ampicillin	>256	>256	>256	0 (0)	0 (0)	50 (100)
Penicillin-G	>64	>64	>64	0 (0)	0 (0)	50 (100)
Piperacillin	>128	>128	>128	0 (0)	0 (0)	50 (100)
Ticarcillin	>128	>128	>128	0 (0)	0 (0)	50 (100)
Amoxicillin-clavulanate	4–> 64	64	>64	7 (14)	3 (6)	40 (80)
Ceftazidime-clavulanate	2–256	16	64	NA	NA	NA
Cefotaxime-clavulanate	0.25–256	16	64	NA	NA	NA
Ticarcillin-clavulanate	32–> 128	128	>128	0 (0)	12 (24)	38 (76)
Piperacillin-tazobactam	4–> 128	64	>128	8 (16)	17 (34)	25 (50)
Cefmetazole	0.25–> 64	2	16	45 (90)	2 (4)	3 (6)
Cefoxitin	2–> 64	8	32	31 (62)	9 (18)	10 (20)
Ceftazidime	8–> 256	64	256	0 (0)	4 (8)	46 (92)
Cefoperazone	≥ 64	>64	>64	0 (0)	0 (0)	50 (100)
Ceftriaxone	> 64	>64	>64	0 (0)	0 (0)	50 (100)
Cefotaxime	32–> 256	>256	>256	0 (0)	0 (0)	50 (100)
Cefepime	64–> 256	>256	>256	0 (0)	0 (0)	50 (100)
Imipenem	0.25–1	1	1	50 (100)	0 (0)	0 (0)
Meropenem	<0.03–0.5	0.06	0.125	50 (100)	0 (0)	0 (0)
Flomoxef **^b^**	0.06–> 64	1	8	45 (90)	3 (6)	2 (4)

^a^ S: susceptible, I: intermediate, R: resistant, and NA: not available (MIC breakpoints are not provided in the CLSI guidelines). ^b^ Susceptibility phenotypes predicted based on the MIC breakpoints for moxalactam (CLSI).

**Table 5 antibiotics-10-00181-t005:** Antimicrobial resistance profiles of *K. pneumoniae* (*n* = 50).

AMR Profile *	No. (%) of Strains
AMP, PCG, PIP, TIC, AMC, TIM, CMZ, FOX, CAZ, CFP, CRO, CTX, FEP	1 (2)
AMP, PCG, PIP, TIC, AMC, TIM, TZP, FOX, CAZ, CFP, CRO, CTX, FEP	4 (8)
AMP, PCG, PIP, TIC, AMC, TIM, FOX, CAZ, CFP, CRO, CTX, FEP	1 (2)
AMP, PCG, PIP, TIC, AMC, TIM, TZP, CAZ, CFP, CRO, CTX, FEP	16 (32)
AMP, PCG, PIP, TIC, AMC, TZP, FOX, CAZ, CFP, CRO, CTX, FEP	2 (4)
AMP, PCG, PIP, TIC, TIM, CMZ, FOX, CAZ, CFP, CRO, CTX, FEP	1 (2)
AMP, PCG, PIP, TIC, TIM, TZP, CMZ, FOX, CFP, CRO, CTX, FEP	1 (2)
AMP, PCG, PIP, TIC, AMC, TIM, CAZ, CFP, CRO, CTX, FEP	8 (16)
AMP, PCG, PIP, TIC, AMC, TIM, TZP, CFP, CRO, CTX, FEP	1 (2)
AMP, PCG, PIP, TIC, TIM, TZP, CAZ, CFP, CRO, CTX, FEP	1 (2)
AMP, PCG, PIP, TIC, AMC, CAZ, CFP, CRO, CTX, FEP	5 (10)
AMP, PCG, PIP, TIC, AMC, TIM, CFP, CRO, CTX, FEP	1 (2)
AMP, PCG, PIP, TIC, TIM, CAZ, CFP, CRO, CTX, FEP	3 (6)
AMP, PCG, PIP, TIC, AMC, CFP, CRO, CTX, FEP	1 (2)
AMP, PCG, PIP, TIC, CAZ, CFP, CRO, CTX, FEP	4 (8)

* AMP: ampicillin, PCG: penicillin-G, PIP: piperacillin, TIC: ticarcillin, AMC: amoxicillin-clavulanate, TIM: ticarcillin-clavulanate, TZP: piperacillin-tazobactam, CMZ: cefmetazole, FOX: cefoxitin, CAZ: ceftazidime, CFP: cefoperazone, CRO: ceftriaxone, CTX: cefotaxime, and FEP: cefepime.

**Table 6 antibiotics-10-00181-t006:** ESBL genes profiles of *E. coli* and *K. pneumoniae.*

ESBL Gene Profiles	*K. pneumoniae*(*n*) (%)	*E. coli*(*n*) (%)
*bla* _CTX-M_ *, bla* _TEM_ *, bla* _SHV_	34 (68)	0 (0)
*bla* _CTX-M_ *, bla* _SHV_	6 (12)	0 (0)
*bla* _CTX-M_ *, bla* _TEM_	3 (6)	17 (34)
*bla* _CTX-M_	1 (2)	12 (24)
*bla* _SHV_	6 (6)	0 (0)
*bla* _TEM_	0 (0)	10 (20)
None	0 (0)	11 (22)

## Data Availability

All data presented in this study are available in this published article.

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
