# Peer review of "In Vitro Efficacy of Flomoxef against Extended-Spectrum Beta-Lactamase-Producing Escherichia coli and Klebsiella pneumoniae Associated with Urinary Tract Infections in Malaysia"

_antibiotics, 2021, doi:10.3390/antibiotics10020181_

Round 1
Reviewer 1 Report
Ngoi et al. evaluated the susceptibility profile and ESBL production of 50 Escherichia coli and 50 Klebsiella pneumoniae isolates that were identified as presumptive ESBL-producers after collection from the urine of patients at the University Malaya Medical Centre. In addition to evaluating the activity of common beta-lactams, the authors also evaluated the susceptibility of the isolate collection to flomoxef, which the authors propose may be a suitable agent for a carbapenem-sparing beta-lactam regimen against ESBL-producing uropathogens. Overall, the methodology is appropriate and I believe the study will be of interest to the readers of Antibiotics; however, I have several comments/suggestions for improving the clarity and presentation of the manuscript:
- The manuscript will benefit from having a native speaker of the English language review the document to fix common grammatical mistakes.
- Line 46 – I don’t think the authors need to parenthetically explain that Escherichia coli and Klebsiella pneumoniae are referred to as coli and K. pneumoniae considering the normal scientific convention is to abbreviate the genus after the first use in the manuscript.
- Lines 49 – 51 – the sentence reads “The majority (> 80 %) of the community-acquired UTIs, also known as uncomplicated UTIs, in otherwise healthy women are caused by uropathogenic coli [UPEC].” I think the sentence can be rephrased to be more clear. For example, “Uncomplicated urinary tract infections are defined as infections in otherwise healthy women in the community, and 80% of such infections are caused by uropathogenic E. coli.”
- As above, the Infectious Diseases Society of America has clinical practice guidelines on uncomplicated UTIs, but the definition of a complicated UTI sometimes varies based on what reference is being used, so it may be helpful to explicitly state the definitions for uncomplicated and complicated UTIs the authors are using in the Introduction.
- In lines 59 – 62, the authors seem to be describing the treatment of uncomplicated cystitis specifically, but only specify that they are discussing “UTIs.” I recommend they explicitly state that the treatment for uncomplicated UTIs is nitrofurantoin, trim-sulfa, fosfomycin, and pivmecillinam (where available, the product is not offered in the United States), and I also recommend that they add the IDSA guidelines on uncomplicated cystitis to the citation they selected.
- In lines 64 – 66, the authors only mention using beta-lactam antibiotics or quinolones in the case of a treatment failure, but beta-lactams and quinolones are often used empirically for complicated urinary tract infections… especially pyelonephritis.
- Can the authors add a couple sentences to the introduction that gives some clinical context to flomoxef to clinician scientists outside of Northeast Asia (apparently the only region where the drug is available)? Basic information like the breadth of antimicrobial coverage, whether it is typically used for empiric or directed therapy, how commonly it is used for UTIs, and why the authors decided to focus on the drug in the current study will be helpful. In the current draft of the manuscript, the reader needs to read all the way to lines 269 – 277 of the Discussion to understand why the authors are interested in investigating that drug specifically.
- Are the antimicrobial combinations of ceftazidime-clavulanate and cefotaxime-clavulanate commercially available? If so, which regions of the world have access to those combination products?
- The copy editor will probably catch a lot of the stylistic edits, but organism names like coli and K. pneumoniae were not italicized through the majority of the manuscript.
- Are the susceptibilities reported in lines 94 – 97 referring specifically to coli isolates that were resistant to 10 or more antimicrobials? If so, can the phrasing of that section be modified to make it clear that the authors are referring specifically to organisms with resistance to 10 or more agents?
- Did the authors exclude bacterial isolates that possessed ESBL enzymes and a carbapenemase enzyme from the study, or did the University of Malaya Medical Centre just happen to collect 100 isolates that produced ESBL enzymes but not a carbapenase? If isolates with carbapenemases were excluded, how often was an isolate identified that possessed an ESBL enzyme and a carbapenemase in comparison to isolates that only possessed ESBL enzymes? It sounds like the authors did not exclude any isolates based on line 248 of the Discussion, but it will be helpful to explicitly mention that no isolates were excluded in the Methods as well. I also recommend explicitly stating the country and city where the hospital is located in the Methods section.
- In lines 186 – 187, the authors sound surprised that cephamycins were not hydrolyzed well by ESBL enzymes, but isn’t that finding consistent with other studies on the substrates of ESBL enzymes? It may be more logical to say something like “Consistent with other studies, the cephamycins remained active against the majority of the investigated isolates” and then site another study for context.
- In Table 6, it was interesting to see the susceptibility of ESBL-producing organisms to non-beta-lactam antibacterials such as nitrofurantoin and trimethoprim-sulfamethoxazole. I would be interested to see their susceptibility to quinolones as well. Is there a specific reason the authors didn’t look at susceptibility to quinolones?
Reviewer 2 Report
GENERAL COMMENTS:
In the present study authors perform susceptibility tests and PCR to detect phenotypic resistances to beta-lactams and beta-lactamase encoding genes in E. coli and K. pneumoniae isolates obtained from patients with UTI. The study, although it is not misread, is not very novel on a methodological level because there have already been other authors in other Asian countries who have investigated the efficacy of flomoxef in vitro in Enterobacteriaceae (doi: 10.1016/j.ijantimicag.2014.11.012. ). In addition, authors have only investigated 4 types of beta-lactamase encoding genes. There are many antimicrobial resistance mechanisms responsible for phenotypic resistances in addition to the 4 types of beta-lactamase encoding genes analyzed by the authors. It cannot be assured that these genes are the only ones responsible for the phenotypic resistances detected. The authors must soften their claims. I would recommend the authors to "sell" their work as a novelty for the region (Malaysia), or as a prevalence study in E. coli/K. pneumoniae isolates from UTI patients in Malaysia. English is not bad but should be reviewed. The authors make grammatical errors (eg: overuse of the article "the").
SPECIFIC COMMENTS:
TITLE
Why authors put the focus on flomoxef? In their objectives is also not clear which the objective is. Maybe authors could add in their title “…in Malaysia”, because flomoxef activity has been already investigated as a treatment against Enterobacteriaceae causing diverse infections.
ABSTRACT
-L13 and throughout: there are grammatical mistakes that should be corrected ( …are two OF the most common…)
-L17: 19 beta-lactam antibiotics used in UTI treatment.
-L25: remained AS the most effective ATBs against… If carbapenems are still in the first position, I am not sure why authors would like to add flomoxef in their title.
INTRODUCTION
Authors give a wide overview on UTIs and their causes, and also exposed the problem with AMR (antimicrobial resistance) in L33 to L75. Right after, they start with their objective (L75-79). However, they do not mention anything about AMR in Malaysia. What is it the current situation in the region? Background is needed on that. Authors could reduce L33-75 and focus on the situation of Malaysia. Also, if authors want to focus on floxemed, they have to justify why.
-L69: please eliminate “is” and “and” and substitute terms such as “requires attention”, which is something obvious but it does not add anything to your text. Same goes for L74.
-L75-79: The objectives should be re-written and be consistent with the title, results and conclusion.
RESULTS
I would recommend first presenting the results in table 2 and then the combinations of resistances in table 1. In table 2, please specify next to each MIC range the outcome (S, for sensitive or R for resistance).
L82: all E. coli isolates (n=x; %)…please specify numbers and/or % in the text too. In this first line authors should reference already to Table 1. Please correct the species names. They should be in italics.
L83-88: when authors use linkers such as however or nonetheless, it seems more a discussion rather than results.
L93: please be aware that you cannot say “the E. coli”. Check the manuscript for these grammar errors.
L103: eliminate “high-level”, it does not sound scientific. Same goes for L129. Authors do not need to list all the ATBs for which a resistance was found. Just refer to table 2.
L151-153: this part should be the first paragraph of your results together with table 6 but the table should be corrected. Authors present here a mix of descriptive data on the patients and on the isolates. This is not correct. Authors should present only demographic data on the patients (age, gender, hospitalization, previous ATB treatment), when available, and this should be on a separated table (new table 1). Also, I would suggest that authors place the AMR frequencies for E. coli and K. pneumoniae in two columns on the right side as part of table 4 and eliminate the data on “simultaneous resistance”, “MIC of flomoxef” and “prevalence of beta-lactamase genes”, since these info can be displayed in the text. Thus, table 6 would not longer exist. Also, in L151, please substitute more than the half by approximately half. Is there any data on or possibility of having mixed infections (E. coli+K. pneumoniae or other bacteria/parasite)?
L153-155: if authors say “proportion”, they should indicate a %. Also, how can authors say, that there is no gender preference? This should be statistically supported. Otherwise, eliminate the statement on gender preference.
L157: most E. coli. Please specify again numbers or %.
L153-161: this is correct in here, but there is no need of a sub-heading. Please eliminate “comparative analysis”.
L164-167: these are methods and should be in methods.
DISCUSSION
L183: an increasing rate is not accurate, this is not a trend over time.
L185: I would suggest: The multiresistance patterns detected in E. coli and K. pneumoniae isolates was accompanied in some cases by ESBL encoding genes.
L189: limitations should be placed in the last paragraph before the conclusion
L194: when do authors recommend the use of flomoxef if carbapenems still do work? The use of flomoxef as an alternative could be the justification of this study and in that case should be place before the objectives.
L200-201: eliminate “concurred” and say “in accordance to other works”. more references are needed for discussion. For instance:
https://pubmed.ncbi.nlm.nih.gov/25600890/
https://pubmed.ncbi.nlm.nih.gov/30658867/
https://www.ncbi.nlm.nih.gov/pmc/articles/PMC7167803/
https://link.springer.com/article/10.1007/s10157-019-01775-w
L202-214: this looks like an introduction and does not add anything to the discussion. Why do authors comment on USA data. Please use some of the links provided above for your discussion. This paragraph is totally disconnected from your results. Authors should compare their prevalence results with other works.
L215: eliminate unanimously
L220: Misuse, overuse, but not heavy use.
L221-226: again, this is for an intro and does not add anything to your manuscript, because you have not investigated plasmids, just 4 genes.
L228-232: There are many mechanisms of acquired AMR resistance not investigated by the authors. For instance, in monobactams beta-lactamases exist but also alteration of penicillin-binding proteins or and decreased permeability are responsible for acquired resistance. Authors should tone down their affirmations in L229 and indicate that since no other mechanism was investigated, other genes or mechanism may contribute to the resistances detected (limitation of the study).
L232: “multiple”??? you only detected 3!
L234: eliminate concurred. You could use: in agreement with other studies…
L236: which sources? The environment can also be a source. Please eliminate “more often than not” and indicate new references.
L242-243: blaOXA is not infrequent. Also, this is a. bit speculative, since authors have not investigated more genes nor other mechanisms. In any case, I would suggest to leave it for the paragraph of limitations, just before conclusions/recommendations.
L246-247: it is not only the betalactamases, as indicated above, but other mechanisms. Please eliminate this sentence.
L249-258: this fits in your introduction, to justify your study.
L256: it will be of interest to describe the Malaysian surveillance system (data collected, since when is active, etc.)
L257: be aware that you have not calculated incidence rates here.
L259: there is no such thing as “safely infer”. Please eliminate.
L262-264: please, place recommendations at the end of the manuscript.
L264-267: what is this info providing to the reader?
L269: a comparison? In this study or in others? Specify please.
L271-284: this looks like background information not linked to your results and it is actually important to justify your study, if your aim is to focus in flomoxef.
L286: statistically correlated? I have not seen any statistical values in the manuscript.
L293-296: it can be due to many other factors such as the ATB use in each country, environmental issues, ATBs used in veterinary medicine, etc. Please, elaborate more on this.
L296-299: and? What is the value of this here? Please elaborate on that. Authors have not investigated the variants here.
L299-302: this is merely speculative.
METHODS
Authors should first explain the patient´s collected data and then talk about the isolates and laboratory methods. I miss statistical analysis. Otherwise authors cannot talk about correlation.
L306: do authors mean one strain per patient? If so, please indicate that instead of saying “non-duplicated”.
L308: what do authors mean by convenience sampling?
L311-314: Please, just simplify it and write, “E. coli and K. pneumoniae isolates were further confirmed by Polimerase Chain Reaction (PCR) through detection of phoA and mdh genes, respectively.” Why did authors choose those conserved genes and no others? Mdh is also present in Klebsiella oxytoca. For E. coli, authors could have chosen uidA gene.
L314: please, abbreviate antimicrobial resistance to AMR. Just write the first time AMR is mentioned in the text the extended form.
L316: authors have to state clearly that those three ATBs were not analyzed by them but data were retrieved from laboratory databases.
L336-348: there is no need to give details if authors followed a published methodology. If modifications from the original protocol were made, just indicate those.
L350: “In this study, ESBL-producing…”
L351: eliminate high-level. You could write: isolates were found to be highly resistant to… or a high proportion (xx%) of E. coli and K. pneumonia isolates were found to be resistant to…
L353: eliminate the sentence
L354: I would suggest: “Carbapenems showed the highest in vitro efficacy to ESBL-producing strains, followed by flomoxef. Therefore, due to the low prevalence of AMR to flomoxef detected here, this ATB can be considered as an alternative to carbapenems in the clinical practice in Malaysia for the treatment of UTIs caused by ESBL-producing E. coli and K. pneumoniae”.
How many Ec and Kp strains were resistant to flomoxef?
Round 2
Reviewer 2 Report
In the current reviewed version, authors have significantly improved their manuscript and it reads very well. In general, the manuscript still needs further revision for minor grammar mistakes, usually related with lack of articles.
Regarding the content, I just have minor comments to add:
-L106-116: the info in this paragraph is not diplayed in the correct order. You should start by introducing table 2, what we observe there and then its results. I would suggest for this paragraph: "Table 2 summarizes both the MIC value and the corresponding phenotype of the ESBL-producing E. coli strains in this study. The minimum inhibitory concentration (MIC) range, median (MIC50) and 90 % efficacy (MIC90) values for all tested β-lactam antibiotics are also shown. Briefly, all E. coli strains (n = 50) were confirmed as ESBL-producers based on Clinical and Laboratory Standards Institute (CLSI) guidelines. High MIC values (MIC50 ≥ 64 μg/mL) were recorded for penicillins, third- and fourth-generation cephalosporins, and two out of five of the β- lactam combination agents when tested against ESBL-producing E. coli strains (Table 2). Carbapenems and flomoxef showed lowest MIC50 values (≤ 0.5 μg/mL). In addition, most of the E. coli strains (n=xx; 82 %) showed simultaneous resistance to at least 10 β-lactam antibiotics (table 3)."
-Table 3 caption: eliminate " isolated from urine specimens" and say: Antimicrobial resistance profiles of the ESBL-producing E. coli strains (n = 50). Same goes for the caption in Table 5.
-L279-281: please, eliminate. It does not add up anything.
